# White Matter Correlates of Domain-Specific Working Memory

**DOI:** 10.3390/brainsci13010019

**Published:** 2022-12-22

**Authors:** Autumn Horne, Junhua Ding, Tatiana T. Schnur, Randi C. Martin

**Affiliations:** 1Department of Psychological Sciences, Rice University, Houston, TX 77005, USA; agh9@rice.edu; 2Department of Neurosurgery and Neuroscience, Baylor College of Medicine, Houston, TX 77030, USA; jhding@mail.bnu.edu.cn (J.D.); tatiana.schnur@bcm.edu (T.T.S.); 3Department of Psychology, University of Edinburgh, Edinburgh EH8 9YL, UK

**Keywords:** working memory, phonological working memory, semantic working memory, diffusion tensor imaging, white matter tracts

## Abstract

Prior evidence suggests domain-specific working memory (WM) buffers for maintaining phonological (i.e., speech sound) and semantic (i.e., meaning) information. The phonological WM buffer’s proposed location is in the left supramarginal gyrus (SMG), whereas semantic WM has been related to the left inferior frontal gyrus (IFG), the middle frontal gyrus (MFG), and the angular gyrus (AG). However, less is known about the white matter correlates of phonological and semantic WM. We tested 45 individuals with left hemisphere brain damage on single word processing, phonological WM, and semantic WM tasks and obtained T1 and diffusion weighted neuroimaging. Virtual dissections were performed for each participants’ arcuate fasciculus (AF), inferior fronto-occipital fasciculus (IFOF), inferior longitudinal fasciculus (ILF), middle longitudinal fasciculus (MLF), and uncinate fasciculus (UF), which connect the proposed domain-specific WM buffers with perceptual or processing regions. The results showed that the left IFOF and the posterior segment of the AF were related to semantic WM performance. Phonological WM was related to both the left ILF and the whole AF. This work informs our understanding of the white matter correlates of WM, especially semantic WM, which has not previously been investigated. In addition, this work helps to adjudicate between theories of verbal WM, providing some evidence for separate pathways supporting phonological and semantic WM.

## 1. Introduction

Working memory (WM)is the cognitive system that allows us to maintain and manipulate information over short time periods [1]. WM supports many other cognitive processes, including understanding [2] and producing [3] language. Here, we report on the neural basis of domain-specific WM, specifically WM for phonological and semantic representations, which are critical for language processing [4]. Previous work has focused on elucidating the gray matter cortical regions supporting WM [5,6,7,8]. Investigating the role of white matter—the myelinated tracts that connect gray matter regions—in supporting WM, has been a more recent endeavor. For a complete picture of the neural basis of WM, it is necessary to investigate the relation between not only WM performance and gray matter cortical regions, but WM and white matter tracts as well. That is, WM cannot be localized to any one brain area. Rather, WM is supported by networks of gray matter regions connected by white matter tracts. Impairments in WM could therefore occur due to the disruption of certain white matter tracts, even if the gray matter regions that they connect are spared. Consequently, relating deficits in domain-specific WM to the integrity of white matter tracts in individuals with brain damage should inform our understanding of how different brain regions communicate with each other to support phonological and semantic WM.

Existing work on white matter correlates of WM has primarily focused on the role of left hemisphere tracts and often does not control for single word processing or gray matter damage when these factors also potentially affect WM performance [9,10,11]. Additionally, the work has focused on measures tapping phonological WM and thus far, no work has investigated the white matter correlates of WM for semantic representations. The present work addresses these gaps in the literature.

### 1.1. The Domain-Specific Model of WM

Understanding the neural network that supports WM can also help adjudicate between theories of WM. There are many different proposals regarding the structure of WM, including embedded processes models—where WM is the activated portion of long-term memory [12]—and buffer models—where WM is supported by a buffer capacity that is separate from long-term knowledge [13,14]. One influential buffer model of WM is the multicomponent model of WM (Figure 1; Baddeley et al., 2021 [13]). The multicomponent model of WM contains one buffer capacity, the phonological loop, for all verbal representations, and another buffer capacity, the visuospatial sketchpad, for visual-spatial representations. More recently, an additional component has been added to the model: the episodic buffer has been proposed to bind together representations from the visuospatial sketchpad, the phonological loop, and episodic long-term memory (LTM) into a cohesive episodic representation [15].

In contrast, the domain-specific model of verbal WM includes separate buffers for phonological (i.e., speech sound), semantic (i.e., meaning), and orthographic (i.e., written) representations (Figure 2; Martin et al., 2021 [14]). The buffers are separate from each other as well as separate from long-term knowledge in their respective domains. The buffer capacities can also be damaged separately from each other. People with brain damage sometimes demonstrate striking double dissociations in their abilities to maintain phonological and semantic information. That is, some people have difficulty maintaining speech sounds but a better ability to retain word meanings after a left hemisphere stroke, suggesting selective damage to the phonological WM buffer. In contrast, others have difficulty maintaining word meanings but a better ability to maintain speech sounds after a left hemisphere stroke, suggesting selective damage to the semantic WM buffer [16,17,18]. The development of the domain-specific model has been heavily influenced by neuropsychological investigations of patients with phonological and semantic WM buffer deficits [14,19,20].

Double dissociations between phonological and semantic WM have been reported. Patients with a prominent deficit for maintaining phonological representations are classified as having a phonological WM deficit, while those who have more trouble maintaining semantic representations are classified as having a semantic WM deficit. Patients with phonological WM deficits do not show standard phonological effects on WM, such as the word length effect and phonological similarity effect. They also perform better with the visual versus auditory presentation of list items, the opposite of the typical performance pattern. Phonological WM is often assessed using a task such as digit matching, where participants hear two lists of digits and must indicate whether the two lists are the same or different. Performance on digit matching relies on maintaining phonological information associated with the list items. Although some role of semantic representations is evident in the digit list recall [21], such tasks primarily tap into phonological WM. Patients with phonological WM deficits perform more poorly on such tasks than on others that focus on semantic information, such as the category probe task [17]. In the category probe task, participants hear a list of words followed by a probe word. Then, they must draw on their short-term retention of the semantic representations associated with the trial items, to indicate whether the probe word is in the same category as any of the list items. In addition to performing worse on the category probe task in comparison to the digit matching task, people with semantic WM deficits do not show an advantage for memory of words over nonwords, though they do show standard phonological effects on WM. In summary, neuropsychological investigations of WM provide evidence for separable phonological and semantic WM buffers. This claim is corroborated by neuroimaging evidence concerning gray matter correlates of domain-specific WM [6].

### 1.2. Gray Matter Correlates of Domain-Specific WM

Most investigations of the gray matter correlates of WM suggest that there are distinct gray matter regions supporting WM capacity in distinct domains. Specifically, the supramarginal gyrus (SMG) has been implicated in phonological WM [6,8,22,23,24] while the inferior and middle frontal gyri (IFG/MFG) and the angular gyrus (AG) have been related to semantic WM [5,6,7,25]. Early work by Paulesu et al. (1993) [23] found that phonological WM was related to activation in the left inferior parietal region, specifically the SMG. They also observed activation in frontal regions, including the IFG, that they interpreted as supporting articulatory rehearsal. The first study to contrast phonological versus semantic maintenance in a functional MRI study was Martin et al. (2003) [22] which reported significantly greater activation in the left SMG for a phonological compared to a semantic maintenance task and marginally greater activation for semantic compared to phonological maintenance in the left IFG/MFG.

Earlier functional MRI findings have been corroborated by more recent work using both univariate and multivariate approaches to functional MRI analysis. Yue and colleagues (2019) [8] observed sustained activation and a load effect in the SMG during the maintenance phase of a phonological WM task. Yue and Martin (2021) [24] offered further evidence for the SMG’s role in phonological WM by using representational similarity analysis to demonstrate that observed patterns of neural activity in the SMG were related to memory items’ phonological similarity, as represented by a theoretical phonological similarity matrix. Neuropsychological work, including lesion-symptom mapping with patients after a tumor resection and neural stimulation during awake neurosurgery, has also supported the SMG’s role in phonological WM [26,27].

Less work has been carried out investigating the gray matter correlates of semantic WM, but the functional MRI work that has been reported with healthy young adults implicates frontal regions, including the inferior and middle frontal gyri. Shivde and Thompson-Schill (2004) [7] found that the short-term maintenance of semantic information was associated with activation in the IFG and MFG, and the maintenance of phonological information was associated with the SMG. Additionally, healthy young adults showed greater activation in the IFG and MFG when participants had to maintain a greater number of semantic representations during a language comprehension task [5].

One neuropsychological investigation into the gray matter correlates of WM that is of particular importance to the proposed work, is that of Martin, Ding, Hamilton, and Schnur (2021) [6], which specifically examined the differences between the neural damage affecting phonological versus semantic WM, using a lesion-symptom mapping approach. The study included 94 patients at the acute stage of a left hemisphere stroke, ruling out the possibility of reorganization of function, and related brain damage to semantic or phonological WM, while controlling for single word processing and the other WM component. Decrements in phonological WM were related to damage in the SMG, as well as to cortical regions in the frontal lobe and subcortical regions, which others have posited are involved in motor aspects of articulation and rehearsal. Decrements in semantic WM were related to damage to the angular gyrus and the inferior frontal gyrus. This recent lesion-symptom mapping study corroborates previous work finding the SMG’s relation to phonological WM and the IFG, MFG, and AG’s relation to semantic WM. Studies of the gray matter correlates of domain-specific WM, therefore, support the hypothesis of distinct gray matter correlates for phonological and semantic domains of WM.

### 1.3. White Matter Correlates of Domain-Specific WM

Compared to studies of the gray matter correlates of domain-specific WM, less is known about the white matter fiber tracts that support WM. Past work has focused on relating white matter tract integrity to individual differences in visuospatial or verbal WM and implicates tracts connecting widespread gray matter cortical regions across all four lobes of the brain [28]. The consensus is that WM relies on communication between disparate gray matter regions that are connected by white matter tracts. In buffer models of WM, such as the multicomponent model [13] and the domain-specific model [14], information must be transferred into the storage buffer (e.g., phonological buffer region) from processing regions (e.g., speech perception regions). The better connected regions involved in processing or storage are, the more information that can be transmitted between cortical regions, thus resulting in increased efficiency of WM processing [29]. It may also be the case that larger or denser axons allow for a greater range of neuronal oscillation frequencies, facilitating communication between brain regions [30]. Miller and Buschman (2015) [31] applied this idea in the WM domain by proposing that, if cognitive functions, such as WM, rely on the synchronous activity of a brain network, then a greater range of possible neuronal oscillation frequencies would facilitate synchronous activity between the regions involved in WM processes.

Work investigating the white matter correlates of verbal WM typically relates white matter integrity to performance on tasks such as letter, digit, word, and nonword span. These tasks depend substantially on the retention of phonological information [32,33,34] and are used to specifically measure the short-term retention of phonological information [17]. While there is some variation across the tasks, the findings on the neural correlates of phonological WM typically implicate frontoparietal tracts, including the superior longitudinal fasciculus, and more specifically, the arcuate fasciculus. This finding has been replicated across many different study populations, including healthy young adults [10,35], older adults [11], children [36], people with a left hemisphere stroke [37], and people with multiple sclerosis [9,38].

For instance, Takeuchi et al. (2011) [35] measured young adults’ WM capacity using a letter span task and found that, after controlling for age, WM capacity was related to white matter volume in the frontoparietal and temporal regions, including regions corresponding to the path of the AF. Further, Burzynska et al. (2011) [10] reported that the integrity of frontoparietal tracts was related to the performance on both the high and low load WM tasks, as well as the level of cortical responsivity (the difference between the BOLD activity in a region for low versus high load conditions) in gray matter regions supporting WM. Furthermore, in a study on middle aged and older adults using tract-based spatial statistics, the left AF was related to performance on WM measures [11]. This finding has been similarly observed in other studies of the white matter correlates of WM in aging [39]. At the opposite end of development, the maturation of frontoparietal white matter tracts, including the AF, has also been related to WM development in children. Ostby and colleagues (2011) [36] measured the radial diffusivity (RD) of the superior longitudinal fasciculus in a group of children and adolescents. RD is a measure of diffusion along the radial plane of the axon and is generally thought to indicate the level of myelination. They found that phonological storage capacity (as measured by forward digit span) was related to the RD of the superior longitudinal fasciculus. The relationship was interpreted as demonstrating the role that white matter myelination, specifically myelination of the superior longitudinal fasciculus, plays in the development of WM.

There is also some evidence that damage to frontoparietal tracts leads to deficits in WM for patients with white matter lesions after a stroke. A patient with selective damage to the superior longitudinal fasciculus and arcuate fasciculus was significantly worse at measures of verbal WM, including forward digit span and word span tasks, compared to the control participants [37]. For patients with multiple sclerosis, a degenerative disease affecting the white matter of the brain, the microstructural degeneration of a diffuse network of white matter tracts connecting the frontal, parietal, and temporal regions, has been observed. This diffuse network included the superior longitudinal fasciculus and the AF and its degradation was associated with WM deficits [9,38].

### 1.4. Implications for Theoretical Models of Working Memory

Investigations of the gray and white matter regions underlying phonological and semantic WM capacities also inform our theories about the structure of WM, including adjudicating between buffer models of WM, such as the multicomponent model of WM [13] and the domain-specific model of WM [14]. While both the multicomponent model of WM and the domain-specific model of WM include buffer capacities for verbal representations, the domain-specific model is unique in that it contains multiple buffers for different types of verbal representations, including phonological and semantic representations. The difference in the level of specificity for the buffers in the two models leads to different predictions about the neural correlates of WM from each model. The multicomponent model of WM contains a phonological loop, a buffer for maintaining phonological representations, but it is unable to explain cases of preserved semantic WM with impaired phonological WM. In the multicomponent model, semantic WM is supposedly supported by the episodic buffer, but the episodic buffer is conceived as a multimodal buffer for the integration of semantic, phonological, and spatial information [15]. Thus, damage to the neural basis of the episodic buffer should affect the maintenance of both semantic and phonological WM. In contrast, the domain-specific model of WM includes separate WM buffers for lexical-semantic and phonological representations, and it predicts that the neural correlates of semantic and phonological WM should be distinct. Our investigation on the white matter correlates of phonological and semantic WM provides a piece of neural evidence that can be used to differentiate between the two approaches to WM and its relation to language processing.

### 1.5. The Current Study

In this work, we used diffusion tensor imaging (DTI) to analyze MRI data from a large group (n = 45) of people who had a left hemisphere stroke. We related the extent of damage to the left hemisphere white matter tracts of interest (Figure 3)—including the arcuate fasciculus (AF), uncinate fasciculus (UF), middle longitudinal fasciculus (MLF), inferior longitudinal fasciculus (ILF), and inferior fronto-occipital fasciculus (IFOF)—to decrements in semantic WM, phonological WM, and language processing abilities. These tracts were chosen based on the literature and/or because they terminate in gray matter regions previously found to support phonological or semantic WM (see further discussion below). In this work, there were two primary aims: (1) replicate and extend past work investigating the white matter correlates of phonological WM and (2) investigate the white matter correlates of semantic WM.

Fronto-parietal white matter regions have been associated with phonological WM performance in studies across many different healthy and clinical populations [11,35,39]. As discussed earlier, one white matter tract which is consistently implicated in phonological WM performance is the AF [40]. The AF is a large bundle of fibers that connects gray matter regions in the frontal, temporal, and parietal lobes. It consists of three subsegments: anterior (or parietal), posterior (or temporal), and direct segments. (See Figure 4.) The anterior and posterior segments of the AF are together referred to as the indirect pathway of the AF. The direct segment lies medial to the indirect pathway. The AF connects the SMG, the proposed location of the phonological WM buffer [6,8], to regions in the frontal lobe that support articulatory rehearsal [41,42] and executive function [43] and regions in the temporal lobe that support speech perception [44]. Thus, we propose that the left posterior segment of the AF may support phonological WM by transferring speech that is perceived in temporal regions to the SMG for maintenance, and then, the left anterior segment passes that information to frontal regions for rehearsal. Our prediction is that the integrity of the posterior and anterior segments of the AF (together, the indirect pathway of the AF) will predict phonological WM performance when controlling for semantic WM, single word processing, and gray matter damage. While past work has implicated the left AF in verbal WM, little of this work has been carried out with people who have experienced a left hemisphere stroke (but rather with healthy children, younger or older adults), and none has controlled for single word processing or semantic WM to understand the relationship between the AF and specifically the maintenance of phonological information. Further, this work is unique in its investigation into the specific roles that the subsections of the AF may play in supporting phonological WM. Past work on the role of frontoparietal white matter in verbal WM has focused on the integrity of the AF as a unitary structure, or even less specifically, the entire superior longitudinal fasciculus (a large white matter bundle that contains the AF, as well as other frontoparietal tracts) [45]. Here we chose to analyze both the AF as a unitary structure, in order to better compare with past work, as well as the individual subsegments of the AF.

To our knowledge, there have been no previous studies investigating the white matter correlates of semantic WM. However, there is evidence that the IFG, a gray matter region in the frontal lobe, is involved in semantic WM [5,6]. Therefore, we predict that the left direct segment of the AF, IFOF, and UF, white matter tracts that connect the IFG with semantic processing regions (Figure 3) [46] will support semantic WM. A recent VLSM study found that, in addition to the IFG, AG damage was also related to semantic WM impairments [6]. Additionally, an RSA investigation into the neural basis of semantic WM reported that during a delay period in a semantic WM task, semantic representations could be decoded from the AG [24]. Thus, we also predict that the left MLF and ILF, which include projections to the AG and connect it to temporal regions supporting semantic knowledge, will also support semantic WM (Figure 3) [47]. Overall, our prediction is that the left UF, MLF, ILF, IFOF, and direct segment of the AF will predict semantic WM after controlling for phonological WM performance, single word processing, and gray matter damage.

## 2. Methodology

### 2.1. Participants

Participants included 45 people with left hemisphere brain damage. Behavioral and imaging data were collected from 24 participants recruited through Rice University, and 21 through Baylor College of Medicine. Participants recruited through Rice University were enrolled in studies in the laboratory between 2005 and 2020. The participants recruited through Baylor College of Medicine were initially recruited as part of a longitudinal study of the effects of a left hemisphere stroke on language, memory, and executive control, from the acute stage to one year post-stroke. All participants had brain damage due to left hemisphere stroke(s) and were at least one year post-stroke at the time of testing. The mean participant age was 60.2 years (SD = 10.9), and the mean education level was 15.3 years (SD = 2.6). Seventeen participants identified as female. The participants recruited through Rice University were tested in accordance with Rice University’s Institutional Review Board. Those recruited through Baylor College of Medicine were tested in accordance with the Institutional Review Board for Baylor College of Medicine.

### 2.2. Neuroimaging Acquisition

Neuroimaging data were collected over many years, and three different scanners were used. (Table 1). The acquisition parameters for the diffusion weighted and the T1 weighted scans associated with each scanner are presented below. While multi-institutional diffusion imaging studies are still uncommon, there is some evidence for the feasibility of combining diffusion weighted data collected across different magnets [48].

Philips Intera 3T acquisition parameters. The acquisition parameters for the participants scanned in a Philips Intera 3T scanner were as follows: (1) Diffusion weighted sequence: TR = 11,098 ms, 70 axial slices, slice thickness = 2 mm, in-plane resolution: 2 mm * 2 mm, 32 directions, b-value = 800; (2) T1 sequence: TR = 8400 ms, 175 sagittal slices, slice thickness = 1 mm, in-plane resolution: 0.94 mm * 0.94 mm.

Philips Ingenia 3T acquisition parameters. The acquisition parameters for the participants scanned in a Philips Ingenia 3T scanner were as follows: (1) Diffusion weighted sequence: TR = 11,676 ms, 70 axial slices, slice thickness = 2 mm, in-plane resolution: 2.2 mm * 2.2 mm, 32 directions, b-value = 800; (2) T1 sequence: TR = 4800 ms, 180 sagittal slices, slice thickness = 1 mm, in-plane resolution: 1 mm * 1 mm.

Siemens Prisma 3T acquisition parameters. The acquisition parameters for the participants scanned in a Siemens Prisma 3T scanner were as follows: (1) Diffusion weighted sequence: TR = 7700 ms, 72 axial slices, slice thickness = 2 mm, in-plane resolution: 2 mm * 2 mm, 64 directions, b-value = 1000; (2) T1 sequence: TR = 2600 ms, 176 sagittal slices, slice thickness = 1 mm, in-plane resolution: 1 mm * 1 mm.

### 2.3. Lesion Tracing

Lesions were identified on T1 or T2 FLAIR (scanned in the axial direction) scans obtained at the same time as the diffusion weighted scans. The resolution of the diffusion weighted and the T1/T2 images was 1 × 1 × 4.5 mm and 0.5 × 0.5 × 4.5 mm, respectively. The diffusion weighted images were registered to T1/T2 images using Analysis of Functional NeuroImages (AFNI; https://afni.nimh.nih.gov/, accessed on 14 December 2022). The lesions were traced on the diffusion weighted images using the Insight Toolkit SNAP (ITK-SNAP; http://www.itksnap.org/pmwiki/pmwiki.php, accessed on 14 December 2022). The images were then normalized into Montreal Neurological Institute (MNI) space using Advanced Normalization Tools (ANTs; https://stnava.github.io/ANTs/, accessed on 14 December 2022; [49].

### 2.4. Tractography and Tract Segmentation

All pre-processing, tractography, and tract segmentation was completed using ExploreDTI [50]. The preprocessing protocol for all diffusion weighted scans included signal drift correction, Gibbs ringing correction, and correction for Eddy currents. The diffusion weighted scans were registered to each participants’ T1 scan to correct for motion and perform the EPI correction.

Whole brain tractography was performed on the processed and registered diffusion weighted data. We used a deterministic tractography approach with the following parameters: (1) FA threshold = 0.2; (2) step length = 1; (3) angle threshold = 30. Whole brain tractography was followed by the virtual dissections of the tracts of interest. The tracts were dissected using hand-drawn regions of interest in each patients’ native space. The AF and it’s anterior, direct, and posterior subsections were dissected manually, based on the methods described by Catani and colleagues (2005) [51]. The IFOF, ILF, and UF, were segmented, based on the methods discussed in [51]. The MLF was segmented, as described in [52]. Fractional anisotropy (FA) values were extracted to quantify the integrity of each tract after segmentation.

### 2.5. Phonological Working Memory (Digit Matching Span)

In the digit matching span task, participants heard two lists of digits presented one after the other. They were asked to respond “yes” if the lists were the same and “no” if the lists were different. The list items were presented at approximately one word per second. The list lengths varied from two to six items. There were six lists for the two-item trials; eight for the three-item trials; six for the four-item trials; eight for the five-item trials; and ten for the six-item trials. Within each set of trials, half of the lists were matching, and half of the lists were non-matching. In the lists that were non-matching, two of the digits presented in the second list were transposed. The position of the transposition was approximately equal across the list positions. The task was discontinued when the participant’s performance dropped below 75 percent correct on a given list length. Linear interpolation between the two list lengths that spanned 75 percent correct was used to calculate the estimated span length. If a participant did not score below 75 percent correct on the longest list length, their span was calculated using linear interpolation, assuming they would have scored 50 percent correct with lists of seven digits.

### 2.6. Semantic Working Memory (Category Probe)

In the category probe task, participants heard a list of words followed by a probe word. They were asked to indicate whether the probe word was in the same category as any of the words from the list. The categories represented in the list items were animals, body parts, clothing, fruit, and kitchen equipment. If the probe word was in the same category as any of the list items, participants responded “yes,” and they responded “no” if the probe word was not in the same category as any of the list items. The responses could be verbal or nonverbal (e.g., pointing or nodding/shaking the head). Prior to administration, the participants were familiarized with the five categories from which the list items were sampled. The list length began with one item and increased up to four items. The items were presented at the speed of approximately one word per second. The category probe span was calculated using linear interpolation as in the digit matching span task described above. If a participant did not fall below 75 percent correct on the longest list length, linear interpolation was calculated, assuming that they would have scored 50 percent correct on the five-item list length.

### 2.7. Single Word Processing (Picture-Word Matching)

In the picture-word matching task, participants saw a black and white line drawing (e.g., a picture of a crown) and were asked a question about the name of the picture [6,53,54]. The name provided could be the target name (e.g., Is this a crown?), a phonological distractor (e.g., Is this a clown?), a semantic distractor (e.g., Is this a hat?), or an unrelated foil (e.g., Is this a knife?). There were a total of 68 trials divided evenly into four presentation sets of 17 items. The participants responded “yes” if the question matched the picture and “no”, otherwise. The responses could be either verbal or nonverbal (e.g., pointing or nodding/shaking the head). The dependent measures from this task were d’ phonological and d’ semantic values that indexed a participant’s ability to discriminate between the target word and either the phonological or semantic distractor, respectively.

### 2.8. Analysis Plan

We used multiple regression to analyze the relationship between tract integrity and WM performance. In our models, integrity of each of our tracts of interest (quantified using FA) was regressed on both phonological and semantic WM, single word semantic and phonological processing, and gray matter damage to the regions where the tract of interest terminates. We chose a multiple regression approach to test our hypotheses because it allowed us to observe the relation between white matter tract integrity and each of our predictor variables independently of the other predictors included in the model. Controlling for gray matter damage in specific WM regions allowed us to test the prediction that white matter tract integrity predicts WM performance beyond damage to gray matter regions that are thought to support WM [55]. For example, the UF connects the IFG, a proposed semantic WM region [5,6,7], with the anterior temporal lobe, a region proposed to represent semantic knowledge [56,57]. Testing the relationship between UF integrity and semantic WM while also controlling for damage specifically to the IFG and anterior temporal lobe would be a strong test of the role that the UF plays in semantic WM. Because phonological and semantic WM are generally correlated with each other as well as single word processing, we made the decision to use white matter tract integrity as our dependent variable in our multiple regression models, regressing it simultaneously on the measures of phonological and semantic WM, single word phonological and semantic processing, and gray matter damage. This allowed us to determine the relation of tract integrity to each of the WM measures while controlling for all other measures.

Because of the extensive brain damage for some participants, some tracts could not be identified, suggesting that in those instances the tract no longer existed. Using an FA value of 0 for such tracts resulted in an approximately bimodal distribution of FA for some tracts (see Appendix A) which resulted in distributions that violated the assumptions of multiple regression—specifically, assumptions of normality of residuals and/or equal variances around the regression line. Thus, we elected not to include the 0 values in the multiple regressions [58]. However, because this resulted in a substantial reduction in sample size for some tracts, we adopted a logistic regression approach for tracts where 10 cases or more were not reconstructed [59]. Specifically, the left hemisphere tracts and the number out of 45 participants that could not be tracked included the AF (18), the anterior segment of the AF (21), the posterior segment of the AF (27), the direct segment of the AF (23), the IFOF (13), and the UF (11). Logistic regression was used to predict the involvement or lack thereof of a tract with both phonological and semantic WM, phonological and semantic single word processing, and gray matter damage to the regions where the tract of interest terminated. For the left AF and its subsections, IFOF, and UF, both logistic and continuous regressions were performed.

In all regression models, we screened for outliers using both studentized residuals and Cook’s d. An observation was considered an outlier if it had a studentized residual of 2.5 or higher or greater than three times the mean Cook’s d value. Outliers were excluded from the multiple regression models predicting the FA values for the left posterior AF and MLF, and no more than two outliers were ever identified and excluded from any model.

## 3. Results

### 3.1. Descriptive Results

The histograms and box plots of the distributions for all white matter tract FA values (both with and without zero values for the untraceable tracts) are presented in Appendix A. The histograms and box plots for the distributions for WM and single word processing measures are presented in Appendix B. The descriptive statistics for all white matter tract FA values are presented in Table 2 and all WM and single word processing measures are presented in Table 3.

### 3.2. Tract Integrity and WM

In all continuous multiple regression models reported here, tract FA was regressed on phonological WM (digit matching), semantic WM (category probe), phonological single-word processing (phonological d’), semantic single-word processing (semantic d’), and the cube root of gray matter damage to the tracts’ termination regions. We transformed the measures of percent damage to gray matter regions by taking the cube root because the distribution of gray matter damage was highly negatively skewed. The predicted relationships between left hemisphere tracts and phonological or semantic WM are outlined in Table 4, in terms of their independent contribution in the multiple regression. The pairwise correlations between the left hemisphere white matter tract FA values and the behavioral measures are presented in Table 5. Using the FDR correction for multiple comparisons (Benjamini and Hochberg, 1995) [60], separately, for the pairwise relations to semantic and phonological WM, phonological WM was related to the whole AF, the direct segment of the AF, the posterior segment of the AF, and the ILF. Semantic WM was related to the direct segment of the AF, the posterior segment of the AF, the IFOF, and the ILF. However, although the pairwise results suggested several relations between tract FA and both phonological and semantic WM, it is important to factor in single-word processing and gray matter damage to terminations because, for example, variations in phonological processing may have reduced pairwise correlations to phonological WM, whereas variations in semantic processing may have contributed to positive correlations. The results of the continuous multiple regression analyses that tested the hypothesized relations between left hemisphere tracts and WM, while including all the control variables, are presented in Table 6. As shown there, two tracts showed significant weights for semantic WM and two for phonological WM. Phonological WM was related to the integrity of the whole AF and the ILF. Semantic WM, on the other hand, was related to the posterior portion of the AF and the IFOF. In regard to correcting for multiple comparisons, the FDR correction cannot be directly applied to the results from several multiple regression analyses. We note, however, that if we treated the 16 total weights for semantic and phonological WM as independent observations, one might have expected that less than one weight would have been significant by chance alone (0.05 × 16 = 0.8) for alpha = 0.05. Thus, the fact that four weights were significant, greatly exceeds this number and strongly suggests that most relations observed here were not due to chance. Additional analyses using logistic regression for the tracts with more than 10 untraceable tracts are presented in the following sections. For all tables, statistical results with *p* < 0.05 are presented in bold.

### 3.3. Arcuate Fasciculus (AF)

Our first prediction was that left AF integrity would be related to phonological WM performance. When we predicted the integrity of the whole left AF FAs using continuous regression, the weight for phonological WM was significant but semantic WM was not. Further, when we predicted the integrity of the posterior subsection of the AF, the weight for semantic WM was significant but phonological WM was not (Table 6). Because there were many untraceable tracts for the AF and its subsections (Table 2) and because prior studies had implicated the AF in phonological WM (e.g., Takeuchi et al., 2011 [35]; Charlton et al., 2010 [39]), we also utilized logistic regression to test the relation between the AF and phonological and semantic WM. We predicted the presence of the AF subsections which have terminations in the SMG, the anterior, and the posterior AF would be related to phonological WM. However, the logistic regression results did not support this prediction (Table 7). Because the direct segment connects temporal lobe semantic regions to frontal regions, we also predicted that semantic WM would be related to the integrity of the direct segment of the AF, but again, the logistic regression results did not support this prediction (Table 7).

### 3.4. Inferior Fronto-Occipital Fasciculus (IFOF)

As predicted, the weight for semantic WM but not phonological WM was significant in the continuous multiple regression model predicting the left IFOF (Table 6). The logistic regression results mirrored the results of the continuous regression in that semantic but not phonological WM predicted the presence of the left IFOF (Table 8).

### 3.5. Inferior Longitudinal Fasciculus (ILF)

When we predicted left ILF FA values, the weight for phonological but not semantic WM was significant (Table 6).

### 3.6. Middle Longitudinal Fasciculus (MLF)

In the model predicting the left MLF FA, neither the weight for phonological nor semantic WM was significant (Table 6).

### 3.7. Uncinate Fasciculus (UF)

We did not observe a significant weight for either WM measure in the multiple regression models predicting the FA for the left UF (Table 6). Because there were many instances where the left UF could not be tracked, we also tested the relation between left UF integrity and WM using logistic regression. We predicted the presence of the UF with both WM measures, single-word processing, and the cube root of damage to UF terminations. Neither semantic nor phonological WM were significant predictors of the UF’s presence (Table 9).

## 4. Discussion

Here, we have reported the relationships between white matter tract integrity and domain-specific WM in a large (N = 45) group of people with left hemisphere brain damage. We predicted that phonological WM would be related to the integrity of the left AF’s anterior and posterior segments. Additionally, we predicted that semantic WM would be related to the integrity of the left direct segment of the AF, IFOF, ILF, MLF, and UF. Our predictions regarding the white matter correlates of phonological and semantic WM were based on the terminations of these tracts. Thus, we predicted that a tract would be involved in phonological WM if it terminated in the SMG and semantic WM if it terminated in the IFG or AG. A summary of the predicted and observed relations between left hemisphere tracts and WM performance is presented in Table 4.

Our predictions for the white matter correlates of phonological WM were partially supported. We reported a relation between the integrity of the whole AF and phonological WM, replicating past work reporting relationships between measures of frontoparietal tract integrity and phonological WM performance [9,10,11]. In addition to the relation between the AF and phonological WM, we also observed an unpredicted relationship between phonological WM and the left ILF. While the ILF connects the temporal lobe with the inferior parietal and occipital lobes, we do not have a detailed understanding of where exactly this tract terminates. While there are certainly distinct patterns, there is also an amount of observed heterogeneity, particularly in brains that have been altered because of brain damage. While we assume the ILF is more often associated with the AG, a semantic WM buffer, it is possible that it has some terminations in the nearby SMG, the proposed phonological WM buffer as well. 

Our predictions for the white matter correlates of semantic WM were also partially supported. The relationship between semantic WM and the IFOF came out as expected. The left IFOF has terminations in frontal regions including the left IFG, which is a proposed semantic WM buffer region [5,6]. We propose that the left IFOF connects gray matter regions in the temporal lobes supporting semantic processing with the IFG, allowing for information in perceptual and semantic processing regions to be transferred to the IFG for semantic maintenance. There is also evidence that, in some people, the IFOF includes terminations in the precuneus region, which includes the AG [61]. Thus, another explanation for the IFOF’s relation to semantic WM could be that it connects two semantic WM regions, the IFG and the AG, as part of a larger network supporting semantic WM. Unexpectedly, we also observed a relationship between the posterior segment of the AF and semantic WM. As with the ILF, we observed heterogeneity in where exactly the posterior segment of the AF terminated in the parietal lobe. Considering the proximity of the SMG (proposed phonological WM buffer) and the AG (proposed semantic WM buffer), it is entirely possible that our method of segmenting each individual patient’s tract in their native space meant that the posterior AF was, in at least a subset of our patients, connecting the left AG with the temporal lobe. 

We did not observe support for the relationships we predicted between semantic WM and the left direct AF, ILF, MLF, or UF in the multiple regression analyses. The direct segment of the AF also has terminations in perceptual processing regions in the temporal lobe, that we predicted would allow it to transfer semantic information from processing regions to the IFG for storage. Similarly, the left ILF and MLF were predicted to support semantic WM because they have terminations in the occipital and inferior parietal lobe, which includes the AG region, as well as the anterior temporal region. In both cases, we predicted that the white matter tracts allow for the semantic knowledge stored in anterior temporal regions to pass to the AG, a semantic WM buffer [56]. What are some possible explanations for why many of the semantic WM predictions were not supported? For the direct AF, ILF, and MLF, it may be that the region of the temporal lobe that these tracts terminate in is not critical for semantic processing across modalities. The hub-and-spoke-model of semantic processing proposes a modality-invariant hub coordinating semantic information across the distributed semantic processing regions [62]. Originally, it was proposed that this hub was located in the anterior temporal lobe [62]. However, more recent evidence has suggested there are gradations within the ATL where modality-invariant semantic processing is related to more middle and inferior portions of the temporal lobe, including the anterior fusiform gyrus [63]. Thus, it may be that while the ILF, MLF, and direct AF all have terminations in the temporal lobe, these terminations may not be in regions supporting modality-invariant processing, which would be most critical for semantic WM. Finally, while we predicted that the UF would be related to semantic WM because it provides a direct connection between the IFG and the anterior temporal lobe, we did not observe a relationship between the UF and WM after accounting for the contribution of other effects using our multiple regression approach. However, while the UF terminates in orbital frontal regions that include an area implicated in aspects of semantic processing (i.e., Brodmann’s area 47; Poldrack et al., 1999) [64], prior studies specific to WM for semantic information have revealed more posterior IFG regions (e.g., Hamilton et al., 2009 [5]).

Our findings about the neural correlates of WM also contribute to our theoretical understanding of WM. Specifically, understanding the neural basis of WM delineates between two buffer models of WM: the multicomponent model of WM and the domain- specific model of WM. The multicomponent model of WM includes a phonological loop which maintains phonological information and an episodic buffer which integrates (and supports the maintenance of) phonological, semantic, and visuospatial information. In contrast, the domain-specific model of WM contains separable buffers for phonological and semantic WM. While the domain-specific WM predicts distinct white matter correlates of phonological and semantic WM, the multicomponent model of WM does not. We did not observe any overlap between the tracts supporting phonological versus semantic WM in our multiple regression analyses. The multicomponent model cannot account for tracts that are only related to semantic WM performance after controlling for phonological WM performance and vice versa. While the domain-specific model of WM contains a buffer specific to semantic WM, the multicomponent model of WM does not. The episodic buffer in the multicomponent model is conceptualized as a capacity for combining phonological, semantic, and visual representations into a cohesive episodic memory. We would expect that if the tracts related to only semantic WM were the neural basis of the episodic buffer, then they should have an independent relation to phonological WM performance as well. Thus, the evidence of neural correlates distinct to semantic WM or phonological WM is most closely aligned with the domain-specific model of WM.

While this work does address many of the limitations of past work on the white matter correlates of WM, it does have its own unique set of limitations that should be addressed in future work. First, a strength of this work was its large sample size, especially for a neuropsychological investigation, but the sample size was achieved by (1) combining neuroimaging and behavioral data from participants recruited from one institution over the course of 15 years and several updates in scanning technology and protocol and (2) adding to that data collected at a different institution and scanning facility. While some past work has suggested that it is feasible to combine diffusion-weighted data collected across multiple institutions in the analyses [48], more recent work has called that claim into question and suggested ways to mitigate the effects of including data collected via different scanners and/or with different scanning protocols [59]. We would note, however, that when a scanner site is simply included as a covariate in the continuous regression models tested here, all of the previously reported significant effects remain significant (*p* = 0.0081–0.035).

## 5. Conclusions

Here, we have reported the white matter correlates of both phonological and semantic WM in a group of participants with left hemisphere brain damage resulting from a stroke. This is the first report of the white matter correlates associated with semantic WM. Our experimental approach controlled for several factors that have been previously unaccounted for in investigations of the neural correlates of WM, including gray matter damage to tract terminations and single-word processing. Finally, this work provides converging neural evidence for the domain-specific model of WM.

## Figures and Tables

**Figure 1 brainsci-13-00019-f001:**
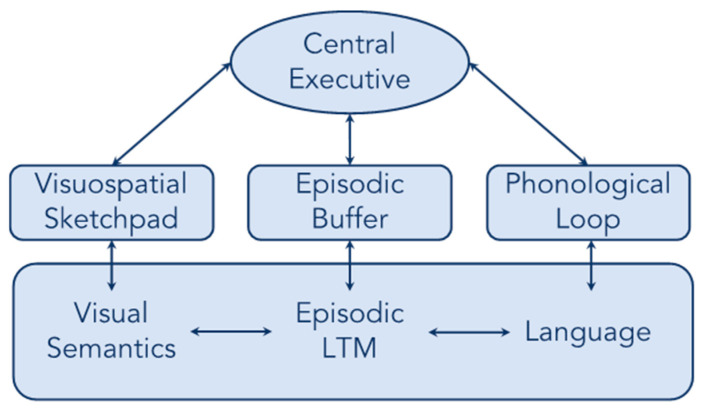
The multicomponent model of WM.

**Figure 2 brainsci-13-00019-f002:**
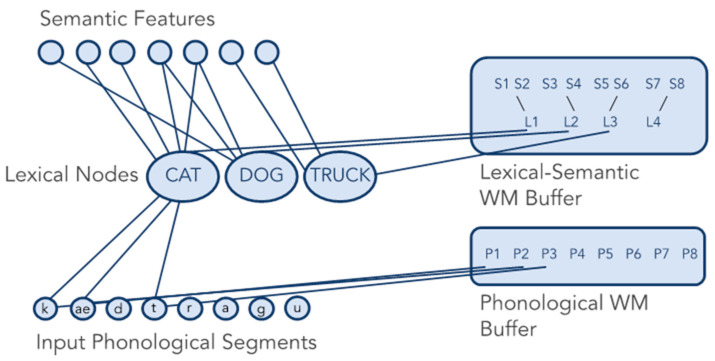
The domain-specific model of WM (the orthographic buffer included in the full version of the model is not pictured here).

**Figure 3 brainsci-13-00019-f003:**
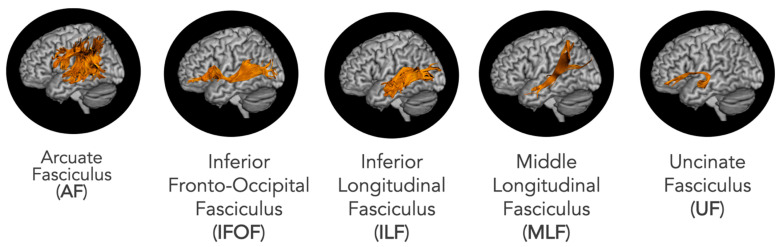
White matter tracts of interest overlaid on three-dimensional renderings of the gray matter cortical regions they connect.

**Figure 4 brainsci-13-00019-f004:**
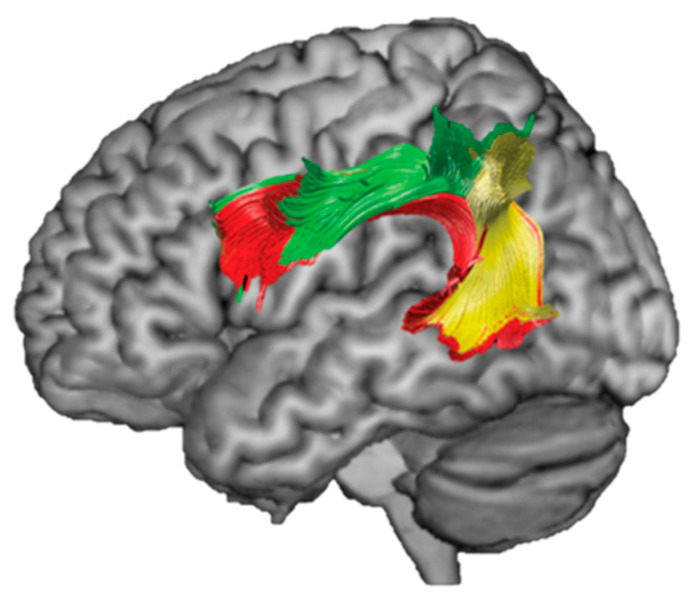
The three subsections of the AF. Red = direct segment. Green = anterior/parietal segment. Yellow = posterior/temporal segment (Adapted with permission from Catani et al. [2005]. Copyright 2004, John Wiley and Sons).

**Table 1 brainsci-13-00019-t001:** Number of participants’ scans acquired by the scanner.

Scanner	Num. Participants Scanned
Philips Intera 3T	18
Philips Ingenia 3T	9
Siemens Prisma 3T	18

**Table 2 brainsci-13-00019-t002:** Descriptive statistics for the white matter tract FA values.

Tract	Number Tracked	Mean FA	SD	Min	Max
Left AF	27	0.24	0.20	0	0.46
Left ant. AF	24	0.21	0.20	0	0.45
Left dir. AF	22	0.20	0.21	0	0.49
Left post. AF	18	0.16	0.20	0	0.45
Left IFOF	32	0.30	0.20	0	0.49
Left ILF	45	0.40	0.045	0.31	0.48
Left MLF	37	0.33	0.16	0	0.48
Left UF	34	0.28	0.16	0	0.42

**Table 3 brainsci-13-00019-t003:** Descriptive statistics for WM and the single word processing measures.

Measure	N	Mean	SD	Min	Max
Digit matching (phonological WM)	43	4.6	1.47	1.76	6.83
Category probe (semantic WM)	43	2.6	1.34	0.45	4.5
Phonological d’ (phonological single word processing)	45	3.4	0.60	1.74	4.11
Semantic d’ (semantic single word processing)	45	2.9	0.66	1.00	3.80

**Table 4 brainsci-13-00019-t004:** Predicted and observed relationships between the left hemisphere tracts and WM.

	Phonological WM	Semantic WM
Left AF	●✓	
Left anterior AF	●	
Left direct AF		●
Left posterior AF	●	✓
Left IFOF		●✓
Left ILF	✓	●
Left MLF		●
Left UF		●

● = Predicted; ✔ = Observed.

**Table 5 brainsci-13-00019-t005:** Pairwise correlations between the left hemisphere white matter tract FAs and the behavioral measures.

Tract	Phonological WM	Semantic WM	Phonological d’	Semantic d’
Left AF n = 27	***r* = 0.49** ***p* = 0.011**	*r* = 0.37 *p* = 0.065	*r* = −0.005 *p* = 0.98	*r* = −0.25 *p* = 0.21
Left anterior AF n = 24	*r* = 0.31 *p* = 0.16	*r =* 0.34 *p* = 0.10	*r =* −0.15 *p* = 0.49	*r =* −0.25 *p* = 0.24
Left direct AF n = 22	***r* = 0.51** ***p* = 0.014**	***r =* 0.51** ***p* = 0.019**	*r =* 0.17 *p* = 0.46	*r =* −0.25 *p* = 0.25
Left posterior AF n = 18	***r* = 0.57** ***p* = 0.014**	***r =* 0.82** ***p* = 0.0001**	*r =* −0.05 *p* = 0.85	*r =* −0.30 *p* = 0.23
Left IFOF n = 32	***r* = 0.37** ***p* = 0.043**	***r* = 0.55** ***p* = 0.001**	*r* = −0.12 *p* = 0.51	*r* = −0.15 *p* = 0.40
Left ILF n = 45	***r* = 0.49** ***p* = 0.0010**	***r* = 0.48** ***p* = 0.001**	*r* = −0.11 *p* = 0.47	*r* = 0.10 *p* = 0.52
Left MLF n = 37	*r* = 0.15 *p* = 0.38	*r* = 0.15 *p* = 0.39	*r* = −0.12 *p* = 0.47	***r* = −0.33** ***p* = 0.046**
Left UF n = 34	*r* = 0.27 *p* = 0.13	*r* = 0.26 *p* = 0.14	*r* = −0.29 *p* = 0.10	*r* = −0.050 *p* = 0.78

**Table 6 brainsci-13-00019-t006:** Results of the continuous multiple regressions predicting the left hemisphere tract FA.

	Phon WM	Sem WM	Phon d’	Sem d’	Gray Matt.
**Left AF**					
Estimate	**0.012**	0.003	−0.005	−0.025	0.012
*t*	**2.41**	0.52	−0.49	−2.01	0.39
*p*	**0.026**	0.61	0.63	0.058	0.70
**Left anterior AF**					
Estimate	0.005	0.008	−0.011	−0.016	0.034
*t*	1.03	1.40	−1.15	−1.28	1.00
*p*	0.32	0.18	0.27	0.22	0.33
**Left posterior AF**					
Estimate	0.004	**0.022**	0.003	−0.009	**−0.20**
*t*	0.80	**4.06**	0.28	−0.95	**−3.00**
*p*	0.44	**0.002**	0.78	0.36	**0.011**
**Left direct AF**					
Estimate	0.013	0.004	0.007	−0.031	−0.030
*t*	1.55	0.43	0.43	−1.64	−0.63
*p*	0.14	0.67	0.68	0.12	0.54
**Left IFOF**					
Estimate	−0.002	**0.023**	−0.013	−0.015	0.010
*t*	−0.32	**2.86**	−1.14	−1.24	0.33
*p*	0.75	**0.009**	0.27	0.228	0.74
**Left ILF**					
Estimate	**0.012**	0.006	−0.021	−0.013	−0.067
*t*	**2.41**	0.93	−1.92	−1.21	−2.69
*p*	**0.021**	0.36	0.063	0.23	0.011
**Left MLF**					
Estimate	0.006	0.003	−0.002	−0.027	0.013
*t*	1.27	0.67	−0.17	−2.63	0.67
*p*	0.22	0.51	0.87	0.014	0.51
**Left UF**					
Estimate	0.007	−0.0001	−0.013	−0.007	−0.043
*t*	1.24	−0.02	−0.95	−0.53	−0.83
*p*	0.23	0.99	0.35	0.60	0.42

**Table 7 brainsci-13-00019-t007:** Logistic regression models predicting the left AF and its subsections.

	Phon WM	Sem WM	Phon d’	Sem d’	Gray Matt.
**Left AF**					
Estimate	0.12	1.07	**−3.03**	0.18	**−8.45**
*χ^2^*	0.14	2.24	**4.56**	0.02	**8.11**
*p*	0.71	0.14	**0.033**	0.88	**0.004**
**Left anterior AF**					
Estimate	−0.26	0.65	**−2.68**	1.99	**−7.42**
*χ^2^*	0.29	1.41	**4.42**	2.33	**6.57**
*p*	0.59	0.24	**0.036**	0.13	**0.010**
**Left posterior AF**					
Estimate	1.11	0.15	0.33	−0.85	−13.83
*χ^2^*	2.8	0.07	0.02	0.43	2.51
*p*	0.094	0.79	0.9	0.51	0.11
**Left direct AF**					
Estimate	0.53	0.84	−1.77	−0.89	**−7.88**
*χ^2^*	1.59	2.54	3.02	0.75	**6.87**
*p*	0.21	0.11	0.082	0.39	**0.009**

**Table 8 brainsci-13-00019-t008:** Logistic regression model predicting the presence of the left IFOF.

	Phon WM	Sem WM	Phon d’	Sem d’	Gray Matt.
Estimate	0.011	**1.27**	−0.29	0.031	0.37
*χ^2^*	0.0	**5.45**	0.14	0.0	0.04
*p*	0.98	**0.02**	0.71	0.97	0.83

**Table 9 brainsci-13-00019-t009:** Logistic regression model predicting the presence of the left UF.

	Phon WM	Sem WM	Phon d’	Sem d’	Gray Matt.
Estimate	−0.43	1.37	2.34	−0.30	**−15.98**
*χ^2^*	0.39	2.25	2.27	0.07	**6.06**
*p*	0.53	0.13	0.13	0.80	**0.014**

## Data Availability

The data presented in this study are available upon request from the corresponding author. The data are not publicly available due to privacy related restrictions.

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
