# Peer review of "White Matter Correlates of Domain-Specific Working Memory"

_brainsci, 2022, doi:10.3390/brainsci13010019_

Round 1

Reviewer 1 Report

I have three points to raise:

1. I don't understand why the authors approach the basic research question by applying linear multiple regression. In my opinion, analyses at the level of network analysis are much more appropriate as tools for the kind of research question the article deals with. But, I don't know to what extent I can ask the authors to change their entire analysis methodology. It's not that multiple regression is necessarily wrong, I just wonder why they chose it. Could they be more explicit with the rationale behind their analyses? 

2. The part about logistic regression is quite hard to understand. A serious issue is that there are large chunks of missing data. Using logistic regressions or any other sigmoidal analysis should be conceptually justifiable. Within the manuscript, the authors do not document why it makes sense on a conceptual level since the variable is not "inherently dichotomous". Simply put, it would be helpful if they could explain their reasoning to the reader beyond merely saying there are gaps in the data and they need to fill them in. I wonder what their results would have been if they had used the participants who had all the data. I also didn't quite understand why there was so much missing data. The authors state that they were not available but do not explain why.

3. Within the manuscript there are missing references that must be corrected.

Reviewer 2 Report

The authors examined associations between phonological and semantic WM and white matter organization in 45 individuals with damages in left hemisphere due to stroke and as measured by FA in 5 tracts of interest (TOI). Results showed associations between FA in left ILF, MLF, IFOF and AF and semantic WM, as well as associations between FA in left ILF and MLF and phonological WM. Authors conlude thatvthe results supports the notion of seperate pathways supports phonological and semantic WM, hence providing support for the domain-specific model of WM.

The study is interesting and on many parameters well-performed. The authors are well informed on state of art for the area of interest, language is well-written, and theoretical approaches are well integrated with the hypothesis and data examined.

However, there are both major and minor considerations and adjustments, which may increase the scientific quality of the study.

Major considerations:

As the sample consists of neurologically inflicted patients and no healthy control group, it should be reflected on whether it is appropiate to generalize the results (support for the domain-specific model of WM). This may be critical, even though authors control for grey matter damage following the stroke, as other studies in white matter has found different associations between white matter metrics and behavioral measures when comparing patients with healthy controls (see i.e. Raikes, A. C., et al. (2018). Diffusion Tensor Imaging (DTI) correlates of self-reported sleep quality and depression following mild traumatic brain injury. Frontiers in Neurology 9, 1–16. doi: 10.3389/fneur.2018.00468).

A thorough discussion of study limitations are lacking. Ie the MRI data was pooled from 2 different sites and 3 different MRI-scanners over at time period of 15 years, where several updates of software would normally be expected. This is a major methodological limitation, for which the authors could have taken some appropiate action to accomodate to, such as controlling / covarying for scanner-type/site/ in the models see i.e Chen et al (2015) Exploration of scanning effects in multi-site structural MRI-studies). It appears that particulary acquisition parameters for the Siemens scanner differed from the 2 Philips scanners, which requires further examination of potential confounding effects. The reference for using multisite scanners (Fox et al 2012) are outdated, since much effort and several studies has been put into investigating these effects in the last decade.

If I understand the stastistical section correct, then primary analyses comprise at least 8 multivariate tests (5 TOIs + 3 segments of one TOI). This is just the TOIs in left hemisphere, subsequently the corresponding TOIs in right hemisphere was explored. Nonetheless, no correction for multiplicity was reported. If the minimum Bonferroni correction for results reported in Table 5 were performed, the corrected p-value for significant results would be 0.05/8=0.00625, in which case the only results which survived would be the association between left posterior AF, IFOF, and ILF (all p values below 0.006). Considering the large number of sensitivity, explorative and post-hoc tests performed subsequently, I would as a minimum expect correction for multiplicity of the primary analyses in order to trust the solidity of results.

This points leads to some further serious remarks: although the authors present an analysis-plan, it is not quite clear from the start that i.e. testing right hemisphere TOIs are explorative (results are also displayed in the same table as left hemisphere). Furthermore, the large number of missing values and as a result of that a change from multiple regression models to logistic regression. Again, a thorough examination and discussion of potential limitations from the missing data (almost half of participants in many TOIs) and the secondary choice of logistic regression are lacking. I.e. what does the logistic regression model really inform us (what is the clinical implications of that the different WM measures predicts presence or abscence of a tract)?

Finally, I somewhat disagree with the conclusion that the results supports the notion of seperate pathways for phonological and semantic WM, and thus the domain-specific model. Apparantly, both phonological and semantic WM were related to left ILF and MLF, whereas only semantic WM were related to  AF and IFOF. This result rather seem to leave the question of either the domain-specific or the multicomponent model unresolved, as the results seem to support both (both shared and seperate pathways). Hence, the conclusion should be more modest and nuanced.

Of minor concerns is the rather unmanageble and unclear reporting of results in way to many tables and figures. It would be very kind to any reader with a more clear presentation, such as collapsing several tables (I.e. all tables for logistic regression in each TOI, such as collapsing Table 6+7+8+9+10+11 to one table, and Table 12+13+14 in to one table etc.). Furthermore, some reporting can be moved to supplements or distilled into essential knowledge / state of arts, which also is the case for the rater extensive introduction, i.e. the section 1.2 on grey matter correlates of domain-specific WM. The current presentation of results appear more messy than actually is the case, and more stringent and condensed use of Tables and figures would ease this impression.
